# No COVID-19 climate silver lining in the US power sector

Max Luke [1✉], Priyanshi Somani[2], Turner Cotterman[3], Dhruv Suri [4] & Stephen J. Lee [5✉]

Recent studies conclude that the global coronavirus (COVID-19) pandemic decreased power sector $CO_2$ emissions globally and in the United States. In this paper, we analyze the statistical significance of $CO_2$ emissions reductions in the U.S. power sector from March through December 2020. We use Gaussian process (GP) regression to assess whether $CO_2$ emissions reductions would have occurred with reasonable probability in the absence of COVID-19 considering uncertainty due to factors unrelated to the pandemic and adjusting for weather, seasonality, and recent emissions trends. We find that monthly $CO_2$ emissions reductions are only statistically significant in April and May 2020 considering hypothesis tests at 5% significance levels. Separately, we consider the potential impact of COVID-19 on coal-fired power plant retirements through 2022. We find that only a small percentage of U.S. coal power plants are at risk of retirement due to a possible COVID-19-related sustained reduction in electricity demand and prices. We observe and anticipate a return to pre-COVID-19 $CO_2$ emissions in the U.S. power sector.

[1] Highland Energy Analytics, Boston, MA, USA. [2] Department of Computer Science and Engineering, Manipal Institute of Technology, Manipal, Karnataka, India. [3] Department of Engineering and Public Policy, Carnegie Mellon University, Pittsburgh, PA, USA. [4] Department of Aeronautical and Automobile Engineering, Manipal Institute of Technology, Manipal, Karnataka, India. [5] Department of Electrical Engineering and Computer Science, Massachusetts Institute of Technology, Cambridge, MA, USA. ✉email: max@highland.energy; leesj@mit.edu

Despite its disruptive impacts on economic and social activities, the global coronavirus (COVID-19) pandemic could have a climate-related silver lining: a sustained reduction in carbon dioxide ($CO_2$) emissions[1–4]. Several studies attribute COVID-19-related decreases in energy demand to decreases in $CO_2$ emissions[1,2,5–10]. Other studies report the impacts of COVID-19 on factors that affect power sector $CO_2$ emissions, such as electricity demand[11–13] and changes to the electricity supply mix[14,15].

The power sector in the contiguous U.S. accounts for 33% of total U.S. $CO_2$ emissions[16]. The power sector is the focus of U.S. decarbonization efforts: since January 1, 2018, ten U.S. states, the District of Columbia, and Puerto Rico enacted legislation that requires reductions in power sector $CO_2$ emissions of 90% or greater in those jurisdictions by 2050 or earlier; while more than 20 electric utilities pledged to reduce their $CO_2$ emissions by 80 −100% by 2050 or earlier[17]. Altogether, those state legislative actions and utility pledges amount to an intended reduction in power sector $CO_2$ emissions of more than 50% by mid-century. The continued retirement of U.S. coal generation capacity constitutes an important component of many states' and utilities' decarbonization efforts[18,19].

Much of the literature on the impacts of COVID-19 on $CO_2$ emissions assumes a causal relationship between the two. Ours is one of the first studies to explore whether such a relationship exists. We do not find evidence that COVID-19 is linked to a reduction in U.S. power sector $CO_2$ emissions except in the two months immediately following shelter-in-place orders, April and May 2020. Nor do we find evidence that COVID-19 is likely to drive out of business more than a small percentage of U.S. coal generation facilities, even though coal-fired power plant retirements are likely to continue for reasons unrelated to COVID-19. We observe returns to pre-COVID-19 levels of $CO_2$ emissions in June 2020, electricity generation in November 2020, and carbon intensity of electricity supply in June 2020.

## Results

**Statistical significance of power sector $CO_2$ emissions.** We show in Fig. 1 deviations from predictive distributions of power sector $CO_2$ emissions generated by a Gaussian process (GP) regression. $CO_2$ emissions are reported in million metric tonnes (MMT) of $CO_2$ on an annualized basis. The black points show historical $CO_2$ emissions values computed using Form EIA-923 and used to fit the GP regression model, and the red points show observed $CO_2$ emissions values for COVID-19-concurrent months. The red

points are not fitted to the GP regression model. The 95% confidence interval (CI) levels are shown by the shaded bars. Blue bars show historical variability in $CO_2$ emissions and red bars show forecasted variability in the counterfactual scenario. The horizontal line within each 95% CI bar shows each GP regression mean (expected) value.

We report the numerical results of the GP regression in Table 1. Column "C" shows the percent deviations in $CO_2$ emissions compared to GP regression mean values and 95% CI bounds, in units of percent deviation, for each month March through December 2020. $CO_2$ emissions values were lower than GP mean values from March through October 2020 and higher in November and December 2020. However, those deviations are outside of predicted 95% CIs in just two months immediately after shelter-in-place orders took effect, April and May 2020. Deviations are not statistically significant for any of the other ten months in the period.

Figure 1 shows that one monthly historical deviation from the GP regression mean, in January 2018, is outside of the 95% CI. In January 2018, the "bomb cyclone" weather event in the Northeast and Mid-Atlantic caused electricity demand to increase and for relatively high-cost oil- and dual-fired generation resources to help meet that demand[20,21].

**Impacts of electricity generation (E) and carbon intensity of electricity supply (C/E) on COVID-19-related $CO_2$ emissions.** Columns "E" and "C/E" in Table 1 show the percent deviations from GP regression mean values and 95% CI bounds for electricity generation (E) and carbon intensity of electricity supply (C/E), respectively, relative to what we would expect in a counterfactual scenario in each month March through December 2020. Figures 2 and 3 show the observed values, GP regressions, and counterfactual values for E and C/E, respectively. Figure 2 shows that the observed values of E are outside of (lower than) the 95% CIs in six months: March, April, May, June, August, and October 2020. The observed values of C/E are outside of (lower than) the 95% CIs in only the two months that followed COVID-19 shelter-in-place orders: April and May 2020 (Fig. 3).

Additional GP regressions are performed to estimate the relative impacts on C/E associated with changes in coal-, natural gas-, and oil-fired electricity generation. The results of those GP regressions are shown in Table 2. The fuel-specific GP regressions exhibit more uncertainty than the regressions related to C, E, and C/E. For example, the forecasted 95% CI for coal generation-related $CO_2$ emissions in April 2020 accounts for +/−24.9% of

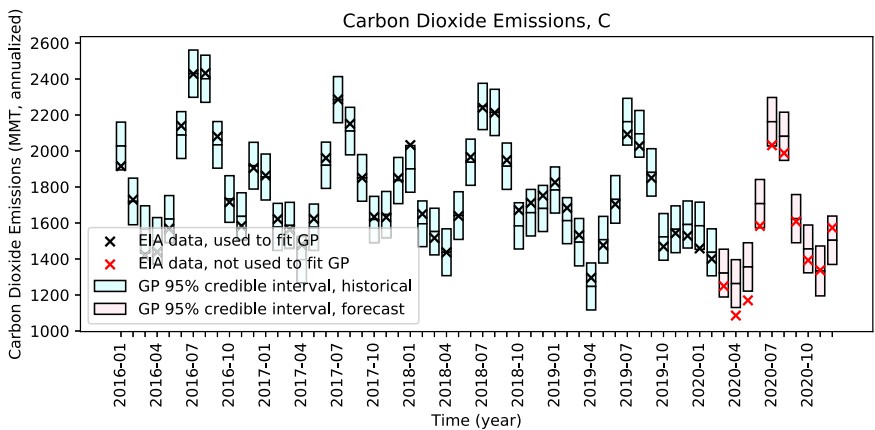

**Fig. 1 Expected and observed $CO_2$ emissions (C).** Gaussian process (GP) regression is used to provide probabilistic estimates of expected $CO_2$ emissions in the absence of COVID-19. The GP 95% confidence interval describes the range in which $CO_2$ emissions are expected 95% of the time, as depicted by rectangular bars for each month. The lines in the middle of the bars denote expected values and the "X" points denote observed values. Observed $CO_2$ emissions are outside of (lower than) the 95% confidence intervals in only two forecasted months: April and May 2020.

| Table 1 Percent deviations between observed values and counterfactual estimates and 95% confidence interval bounds for $CO_2$ emissions (*C*), electricity generation (*E*), and carbon intensity of electricity supply (*C/E*) in the U.S. power sector. | | | |
|---|---|---|---|
| | *C* | *E* | *C/E* |
| March 2020 percent deviation (95% confidence interval bounds) | −5.5% (+/−10.0%) | −3.3% (+/−2.5%) | −2.8% (+/−7.0%) |
| April 2020 percent deviation (95% confidence interval bounds) | −14.0% (+/−10.5%) | −8.8% (+/−2.5%) | −8.6% (+/−7.1%) |
| May 2020 percent deviation (95% confidence interval bounds) | −13.7% (+/−9.9%) | −5.7% (+/−2.5%) | −9.4% (+/−7.1) |
| June 2020 percent deviation (95% confidence interval bounds) | −7.3% (+/−7.9%) | −2.4% (+/−2.1%) | −5.5% (+/−6.4) |
| July 2020 percent deviation (95% confidence interval bounds) | −6.1% (+/− 6.2%) | −1.7% (+/−1.9%) | −2.8% (+/−5.9%) |
| August 2020 percent deviation (95% confidence interval bounds) | −4.5% (+/−6.4%) | −2.7% (+/−1.9%) | −0.6% (+/−6.0%) |
| September 2020 percent deviation (95% confidence interval bounds) | −1.0% (+/−8.2%) | −1.4% (+/−2.3%) | 0.9% (+/−6.4%) |
| October 2020 percent deviation (95% confidence interval bounds) | −4.3% (+/−9.1%) | −3.8% (+/−2.4%) | −0.7% (+/−6.7%) |
| November 2020 percent deviation (95% confidence interval bounds) | 0.3% (+/−10.4%) | −0.6% (+/−2.6%) | −1.3% (+/−7.1%) |
| December 2020 percent deviation (95% confidence interval bounds) | 4.6% (+/−8.9%) | 1.6% (+/−2.3%) | 2.8% (+/−6.6%) |
| Average deviation, March through December 2020 | −5.1% | −2.9% | −2.8% |

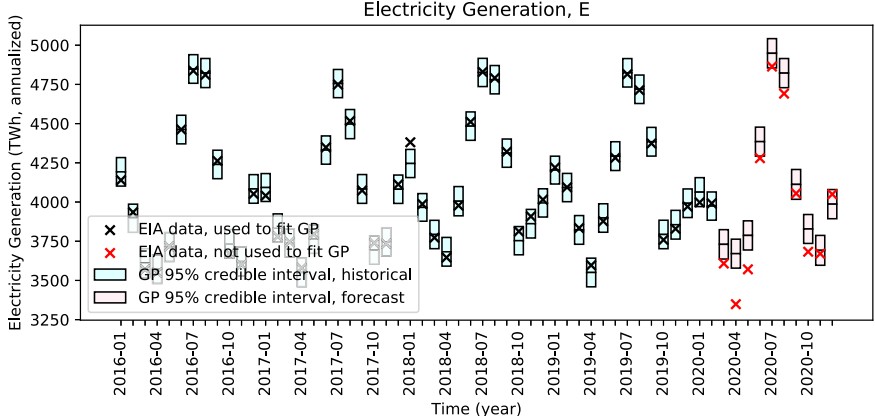

**Fig. 2 Expected and observed electricity generation (*E*).** Observed electricity generation is outside of (lower than) the 95% confidence intervals in six forecasted months: March, April, May, June, August, and October 2020.

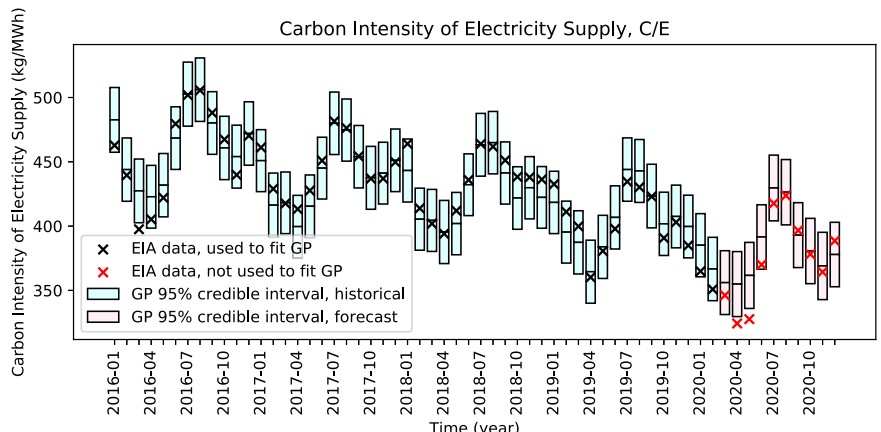

**Fig. 3 Expected and observed carbon intensity of electricity supply (*C/E*).** Observed carbon intensity of electricity supply is outside of (lower than) the 95% confidence intervals in two forecasted months: April and May 2020.

| Table 2 Average counterfactual and observed power sector $CO_2$ emissions (*C*) attributable to coal, natural gas, and oil, March to December 2020, contiguous U.S.; and average percent deviations between counterfactual and observed emissions for each fuel source. | | | |
|---|---|---|---|
| | *C* from coal | *C* from natural gas | *C* from oil |
| Counterfactual average emissions, March−December 2020 (MMT, annualized) | 853.5 | 674.7 | 9.8 |
| Observed average emissions, March−December 2020 (MMT, annualized) | 780.1 | 664.6 | 10.7 |
| Average percent deviation, March−December 2020 | −8.6% | −2.0% | 12.7% |

the mean forecasted value. This implies that that historical fuel-specific $CO_2$ emissions, heating degree day (HDD), and cooling degree day (CDD) data are less predictive of future fuel-specific $CO_2$ emissions using GP regressions than the same for total $CO_2$ emissions, $E$, or $C/E$. Nevertheless, we find that the average reductions in $CO_2$ emissions from coal- and natural gas-fired electricity generation are 8.6% and 2.0%, compared to an average increase in the oil-fired generation of 12.7%.

**Impact of COVID-19 on U.S. coal plant retirements**. We analyze the expected profitability from March 2020 to December 2022 of the 845 coal-fired electricity generation units operating in the seven unbundled power market regions in the United States. Ninety of the 845 coal-fired generation units, representing only 2.8 GW or 1.9% of coal generation capacity in the seven power market regions, were expected to be profitable prior to COVID-19 but are no longer expected to be profitable due to COVID-19-related reductions in electricity prices. We consider those 90 units to be at risk of early retirement due to COVID-19. The mean age and operating capacity of those 90 units are 59.1 years and 31.1 MW, respectively. The newest units were operational in November 2012 and the oldest unit was operational in July 1954. A single at-risk unit resides in the Southwest Power Pool (SPP) market region, while the remaining units reside in the Mid-continent Independent System Operator (MISO) market regions: 79 units in MISO Zone 6, nine units in MISO Zone 4, and one unit in MISO Zone 9.

Estimates of the impacts of COVID-19 on the monthly profitability of coal-fired electricity generation units in aggregate are illustrated in Fig. 4. Figure 4a shows estimates in a

counterfactual scenario in which COVID-19 had not occurred, and Fig. 4b shows estimates in a scenario that reflects our current expectations. Squares represent monthly revenues, diamonds represent monthly costs, and the shaded areas represent monthly profits or losses. Coal generation units in total are $6.5 billion less profitable in the current expectations scenario relative to the counterfactual scenario over the entire period, in present value terms. Of that amount, coal generation units earn $4.5 billion less profit in the current expectations scenario in March through December 2020.

## Discussion

Multiple studies conclude that COVID-19 is responsible for reductions in U.S. power sector $CO_2$ emissions[2,5,9,10] and others conclude that the reduction in $CO_2$ emissions or electricity demand is permanent[3,4,13,22]. For example, researchers conclude in a recent report[13] that COVID-19-related shelter-in-place orders could trigger a sustained long-term reduction in U.S. electricity demand of 65−160 TWh or 1.6−4.0% of annual electricity demand. In their central scenario, the authors estimate a transition of about 11% of U.S. office workers to permanent work-from-home positions, permanent decreases in office and retail-related electricity consumption, and a permanent increase in residential electricity consumption.

We find little evidence of permanent reductions in our analysis of observed changes in U.S. power sector $CO_2$ emissions in the context of normal historical variability. We report statistically significant reductions in $CO_2$ emissions ($C$) in only April and May 2020, and thereafter a return to the levels of $CO_2$ emissions that we would expect in the absence of COVID-19.

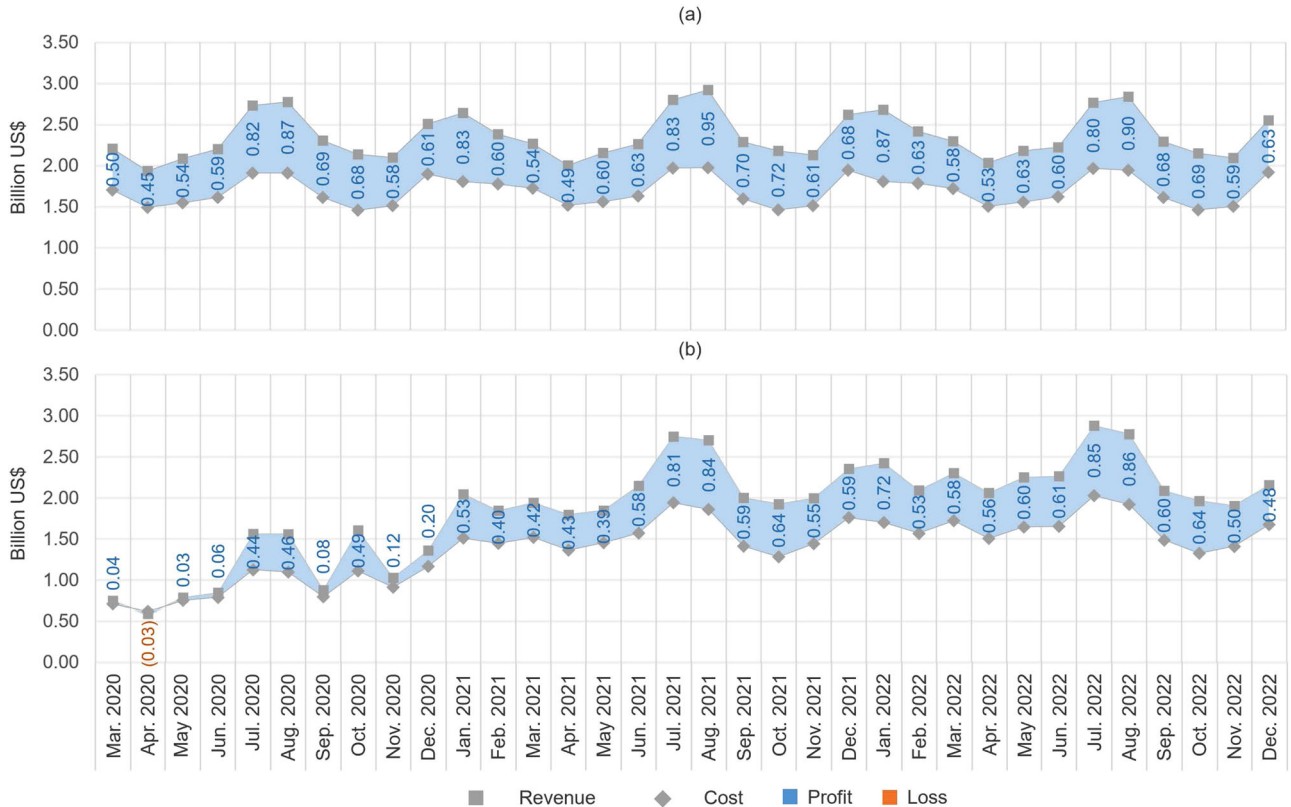

**Fig. 4 Monthly coal generator revenues, costs, and profits, March 2020 through December 2022.** Monthly revenues, costs, and profits for all coal plants in aggregate are shown in the counterfactual scenario in which COVID-19 had not occurred (panel a) and the current expectations scenario (panel b). Squares represent revenues, diamonds represent costs, and the shaded areas represent profits or losses.

With respect to electricity generation (E) and carbon intensity of electricity supply (C/E)—two factors that bear on power sector $CO_2$ emissions—we report returns to levels that we would have expected in the absence of COVID-19 (Fig. 1).

For E, we observe statistically significant reductions in March, April, May, June, August, and October 2020 compared to what we would have expected in absence of COVID-19, but observe values within typical ranges in November and December 2020 (Fig. 2 and Table 1).

Qualitatively, we expect E to remain in typical ranges as economic activity in the U.S. returns to pre-COVID-19 levels. While the U.S. Bureau of Economic Analysis estimates that U.S. real gross domestic product (GDP) declined 9.0% in the second quarter of 2020 relative to real GDP in those quarters in 2019[23], GDP in the third and fourth quarters was only down 2.8%, and 2.5%. A first COVID-19 economic relief bill was signed into law by President Trump on March 27, 2020 and a second by President Biden on March 11, 2021. Owing to these developments and to the expected continued distribution of effective COVID-19 vaccinations, the Board of Governors of the Federal Reserve System, the U.S. Congressional Budget Office, the Organization for Economic Co-operation and Development, and other authorities anticipate a return to pre-COVID-19 economic growth rates in 2021[24–28].

With respect to C/E, we observe statistically significant reductions in April and May 2020 and a subsequent return to levels that we would expect in the absence of COVID-19 (Fig. 3 and Table 1). Nonetheless, we show an average reduction in $CO_2$ emissions associated with coal-fired electricity generation of 8.6% compared to an average reduction in natural gas-fired $CO_2$ emissions of only 2.0% (Table 2). We explore the possibility of COVID-19-related coal power plant retirements and find that COVID-19 is likely to put at risk of retirement less than 2% of coal generation capacity in the contiguous U.S. through 2022.

## Methods

**Statistical significance of power sector $CO_2$ emissions.** Our analysis of U.S. power sector $CO_2$ emissions requires three types of data: net generation by fuel, emissions factors by fuel, and heating and CDDs. We obtain net generation by fuel from the U.S. Energy Information Administration (EIA) Form EIA-923[29], which reports monthly generation and fuel consumption for every power plant. We obtain Form EIA-923 data for the period between January 2016 and December 2020. Monthly plant-level emissions are computed by multiplying fuel consumption for each plant by fuel code-specific emissions factors published by the EIA and U.S. Environmental Protection Agency (EPA)[30–32]. Total $CO_2$ emissions for the contiguous U.S. are evaluated by aggregating plant-level emissions. We obtain population-weighted heating degree day (HDD) and cooling degree day (CDD) data from the EIA[33].

We employ a Gaussian process (GP) regression model to forecast $CO_2$ emissions in the contiguous United States. GP regression models are a class of Bayesian nonparametric models. They assume that every finite collection of random variables has a multivariate normal distribution. GP regressions can be used, as in our case, to forecast the likely ranges of variables corresponding to future periods based on observed historical data. GP regressions are appropriate for testing the statistical significance and estimating the uncertainty of outcomes impacted by seasons and other periodic variables[34,35]. GP regressions have been used to forecast electricity demand and prices, wind and solar power generation, $CO_2$ emissions, battery state-of-charge, and optimal grid management strategies, among other related applications[36–44].

We develop a GP regression that represents a counterfactual scenario in which the COVID-19 pandemic had not occurred. It is developed using the Python GPy library[35]. The GP regression is fit on vector time series data that describes $CO_2$ emissions, HDDs, and CDDs in the contiguous U.S. To forecast $CO_2$ emissions, we fit a constant term (bias kernel), a trend term (linear kernel), and a combination of sinusoidal terms (a standard periodic kernel) to historical $CO_2$ emissions data. The constant and trend terms capture average year-over-year trends while the periodic term, which is constrained to model one-year periods, describes repeating seasonal patterns in the time series. Bias and linear kernels are used to model the impacts of population-weighted HDDs and CDDs on $CO_2$ emissions. The GP model is fit using a SciPy implementation of the L-BFGS-B algorithm with five random restarts[45]. The L-BFGS-B algorithm is a standard optimization technique[46,47]; L-BFGS-B and multiple random restarts are commonly used to fit GPs. Our

optimization settings yield model runs that are highly stable and reproducible. The GP model explicitly captures weather, medium-term $CO_2$ emissions trends, and seasonality. We attribute the remaining Gaussian noise to factors that are not modeled explicitly, such as short-term macroeconomic changes, discrete or non-linear changes to the power-system, and outlier events.

The GP regression is fit to historical emissions data for the period January 2016 through February 2020, the last month before COVID-19-related shelter-in-place orders took effect. This historical period is long enough to obtain a strong model fit but not so long as to necessitate additional nonlinear approximations or more complicated multiyear regression methods. The regression model is used to forecast emissions from March through December 2020. Such a forecast is a probabilistic representation of what emissions would have been in a counterfactual scenario in which COVID-19 had not occurred.

We compare observed to forecasted data and evaluate deviations between the two. The GP methodology allows us to estimate the statistical significance of these deviations. The model generates Gaussian distributions of values in each forecasted month in the counterfactual scenario. Given these Gaussian distributions, statistical hypothesis testing amounts to determinations of 95% CI levels and a simple decision rule: if the observed $CO_2$ emissions level is outside of the 95% CI, we reject the null hypothesis that the observation could have occurred with reasonable probability in the absence of COVID-19. In other words, if the observed $CO_2$ emissions level is outside of the 95% CI, we infer statistically significant impacts of COVID-19 on $CO_2$ emissions. If the observed value is within the 95% CI, we accept or fail to reject the null hypothesis and conclude that deviations are not statistically significant under our decision rule.

**Impacts of electricity generation (E) and carbon intensity of electricity supply (C/E) on COVID-19-related $CO_2$ emissions.** We apply the same steps and use the same data sources to derive time series' for electricity generation (E) and carbon intensity of electricity supply (C/E) that we do to derive a time series for $CO_2$ emissions.

We assess counterfactual and observed values of E and C/E. The counterfactual scenario assumes the continuation of historical power sector $CO_2$ emissions in the absence of COVID-19. The same probabilistic modeling framework that is described in the previous subsection is used to estimate counterfactual values for E and C/E.

As in our counterfactual analysis of $CO_2$ emissions, we include a linear kernel, a bias kernel, and a constrained standard periodic kernel in the GP regressions to capture both short-term and medium-term trends in E and C/E over a period from January 2016 to February 2020. Linear and bias kernels are also used for HDD and CDD data. The GP regression-defined counterfactual data is compared to observed data from March to December 2020.

E and C/E are modeled independently and the joint relationships between those terms are not modeled. As such, the product of the E and C/E values in a given month is not necessarily equal to C, the $CO_2$ emissions that are computed in that month.

**Impact of COVID-19 on U.S. coal plant retirements.** Estimates of coal-fired power plant profitability rely on data related to electricity market prices and capacity auction clearing prices. We also rely on data related to the locations, installed generating capacities, variable costs, and fixed costs of coal-fired electricity generation units.

Historical hourly zonal electricity market prices are obtained from S&P Global Market Intelligence (S&P)[48]. We obtain hourly historical price data for 57 electricity market zones for the period between January 1, 2018 and December 31, 2020.

Forecasts of monthly average regional electricity market prices are obtained from the U.S. Energy Information Administration (EIA)[49,50]. The EIA publishes monthly average wholesale electricity market prices in a single zone in each electricity market region. Two market price forecasts published by the EIA are obtained: a forecast published in January 2020 prior to COVID-19-related shelter-in-place orders, and another forecast published in January 2021, after shelter-in-place orders took effect.

Two electricity market price scenarios are constructed, hourly for each zone, from the monthly EIA forecasts. A counterfactual hourly price scenario reflects the electricity market price forecasts from March 2020 through December 2022 published by the EIA in January 2020. The other hourly scenario reflects our current expectations of electricity market prices. The current expectations scenario reflects actual historical market prices from March through December 2020 and electricity market price forecasts from January 2021 through December 2022. Figure 5 shows the capacity-weighted monthly average electricity prices in our counterfactual and current expectations scenarios.

Capacity auction clearing prices are obtained from S&P for the four electricity market regions that administer forward capacity auctions: Midcontinent Independent System Operator (MISO), New England, New York, and Pennsylvania−New Jersey−Maryland (PJM) interconnection[51]. In these four regions, generators submit offers to electricity system operators to provide generation capacity in a future capacity commitment period, in exchange for payment from electricity system operators. MISO, New England, and PJM run annual capacity auctions for forward capacity commitment periods beginning June 1 and ending May 31. New

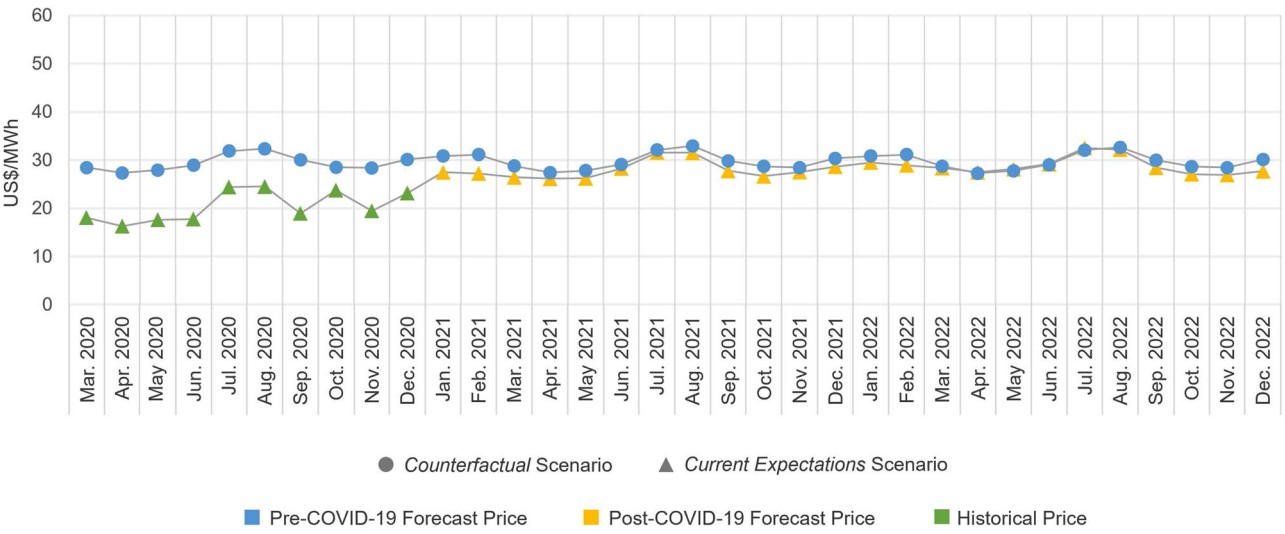

**Fig. 5 Generation capacity-weighted monthly average electricity market prices, counterfactual scenario and current expectations scenario.** We show monthly average prices across all electricity market regions weighted by the coal generation capacity in each region. Circles show counterfactual estimates and triangles show estimates under current expectations. Blue data points reflect counterfactual price forecasts. Yellow data points reflect current expectations price forecasts. Green data points reflect actual historical prices.

York runs biannual capacity auctions that correspond to a winter capacity commitment period between November 1 and April 30, and a summer capacity commitment period between May 1 and October 31. Capacity auction clearing prices are established on a zonal basis.

We obtain actual zonal capacity auction clearing prices through May 31, 2023 for New England. For PJM, price data are available through May 31, 2022. We assume that prices for the subsequent annual commitment period, ending May 31, 2023, are the averages of the prices in the previous five periods. For MISO, price data are available through May 31, 2021. We assume that prices for the subsequent annual commitment periods, ending May 31, 2022 and 2023, are the averages of the prices in the previous four periods. For New York, price data is available through April 30, 2021. We assume that prices for the two subsequent summer commitment periods and the two subsequent winter periods are the averages of the prices in the previous three summer and winter periods, respectively.

The locations, installed generation capacities, and variable costs of each coal-fired electricity generation unit in the seven U.S. electricity market regions are obtained from S&P[52]. Such data is obtained for 2019, the latest year in which data is available. Those data report the regional and zonal locations of each generation unit, the month and year that each unit entered into service, the operating capacity of each unit, and the variable and fixed costs of each unit. Coal-fired cogeneration facilities that produce both electricity and heat are excluded. Such facilities typically supply electricity directly to industrial and commercial facilities. It is difficult to estimate the profitability of such units because they do not earn electricity market revenues for the electricity they supply directly to those facilities.

Finally, an estimate of the weighted average cost of capital (WACC) of coal-fired electricity generation units is obtained from the U.S. National Renewable Energy Laboratory (NREL) Annual Technology Baseline 2019[53]. We adopt an annual WACC of 4.61%.

The profitability of coal-fired power plant units is estimated. The profitability ($P$) for a given month ($m$) and generation unit ($u$) is calculated as that unit's electricity and capacity market revenues ($E_{m,u}$ and $C_{m,u}$, respectively) net of variable operating costs and fixed operating and maintenance (O&M) costs ($V_{m,u}$ and $F_{m,u}$ respectively):

$$P_{m,u} = (E_{m,u} + C_{m,u}) - (V_{m,u} + F_{m,u}) \qquad (1)$$

We describe below the methods we use to estimate expected zonal hourly electricity market prices, monthly electricity market revenues and variable costs, monthly capacity market revenues, and overall profitability for the period between March 1, 2020 and December 31, 2022.

*Zonal hourly electricity market prices.* Profitability is estimated across two sets of electricity market prices: one set based on an electricity market price forecast published by the EIA in January 2020, prior to COVID-19-related shelter-in-place orders, and another set based on actual historical prices between March and December 2020 and an electricity market price forecast published by the EIA in January 2021.

Each of the seven electricity market regions has multiple electricity market zones. The EIA publishes average monthly market price forecasts for each of the seven electricity market regions. Those monthly regional market price forecasts are converted to hourly zonal market price forecasts. Such a step is necessary to

accurately model expected electricity market revenues, variable costs, and capacity market revenues, which are allocated on a zonal basis. The following procedure is performed to determine the forecasted hourly prices in a given month and electricity market zone:

1. For a given month and electricity market zone and region (e.g., June, American Electric Power zone, PJM region), average prices in that month (e.g., June) are determined for each of the three years prior to 2021 for which we obtain historical data, 2018, 2019, and 2020.
2. We compute the differences of those historical average prices with the corresponding monthly price forecast in the appropriate electricity market region (e.g., June 2020, PJM region) published by the EIA.
3. The hourly prices from the historical month associated with the smallest difference that we compute in Step 2 are adopted as our hourly zonal forecast price profile.
4. We shift the hourly zonal forecast price profile up or down by a constant value such that the average monthly zonal price (e.g., June 2020, American Electric Power zone) is equal to the average monthly regional price (e.g., June 2020, PJM region) published by the EIA.

That series of steps are applied to each forecast month and each zone in each electricity market region. Two sets of hourly price forecasts are developed for each zone: one set based on monthly price forecasts published by the EIA in January 2020, prior to COVID-19-related shelter-in-place orders, and another set based on actual historical prices between March and December 2020, and monthly price forecasts published by the EIA in January 2021.

*Monthly electricity market profits and variable costs.* The electricity market revenue and variable operating costs for a given generation unit are estimated by determining the number of hours in a month in which that unit is online. We assume a generation unit is online for the hours in which the electricity market price is greater than or equal to the unit's variable cost of operation. For each month, we generate a price duration curve (PDC) in which we order hourly electricity prices from highest to lowest. The PDC is used to estimate total variable operating costs and electricity market revenues.

The use of PDCs is illustrated in Fig. 6. In Fig. 6, we show two PDCs that correspond to hourly electricity prices in June 2020 in the American Electric Power (AEP) zone in the PJM region. The prices in Fig. 6a reflect an electricity price forecast published by the EIA in January 2020, prior to shelter-in-place orders, and the prices in Fig. 6b reflect actual prices in June 2020. The horizontal dashed lines reflect the variable operating cost ($/MWh) of a coal unit in the AEP zone, "Rockport ST1." Monthly electricity market profits are shown in blue and monthly total variable operating costs ($) in yellow. The vertical gray lines indicate the numbers of hours in which it is profitable for Rockport ST1 to operate in each of the two scenarios.

Electricity market profits and variable operating costs are calculated for each of the 845 coal-fired generation units in the seven electricity market regions in the U.S.

*Monthly capacity market revenues.* Four of the seven electricity market regions, MISO, New England, New York, and PJM, run annual or bi-annual generation capacity auctions in which those regional operators solicit bids from generation units to be available to provide capacity in a future capacity commitment period[51].

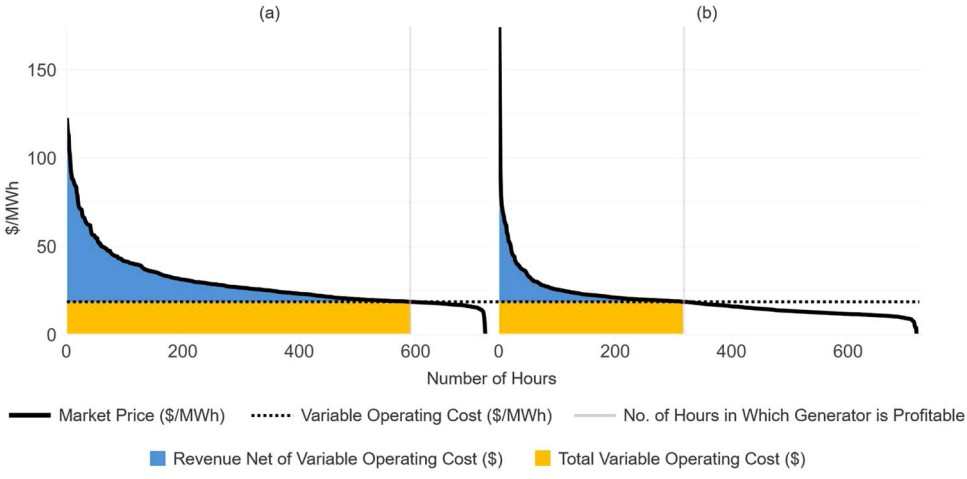

**Fig. 6 Monthly price duration curve, electricity market revenues, and variable operating costs using a price forecast published prior to shelter-in-place orders, and actual prices.** Electricity market profits (blue area) and total variable operating costs (yellow area) are shown for "Rockport ST1", a coal-fired generation unit in the American Electric Power (AEP) Zone in the Pennsylvania—New Jersey—Maryland (PJM) interconnection electricity market, in the month of June 2020, using a price forecast published prior to shelter-in-place orders (panel a) and actual prices (panel b).

We assume that a generation unit bids into a capacity auction such that the unit can expect to be profitable if its bid clears in a given capacity commitment period. For a generation unit ($u$) that does not otherwise expect to be profitable in a given capacity commitment period (cp), that unit submits a capacity bid such that the present value of cash flows associated with its bid ($B_{cp,u}$) would cover the present value of the sum of its variable costs and fixed O&M costs ($V_{cp,u}$ and $F_{cp,u}$ respectively) net of the present value of electricity market revenues ($E_{cp,u}$). For a generation unit that does expect to be profitable in a given capacity commitment period, that unit bids zero dollars into the capacity auction for that period. Equation (2) shows the bidding behavior for generation units that do not expect to be profitable in electricity markets alone, in a given capacity commitment period.

$$B_{cp,u} = V_{cp,u} + F_{cp,u} - E_{cp,u} \quad (2)$$

Capacity auction clearing prices are obtained or estimated for each capacity commitment period through 2022 for each of the four regions that run forward capacity auctions. Those clearing prices are established on a zonal basis. A generation unit that submits a bid that is equal to or lower than the auction clearing price receives capacity market revenues. A generation unit ($u$) that clears in a given commitment period (cp) in a given zone ($z$) receives capacity market revenue ($C$) in a given month ($m$) equal to the zonal auction clearing price ($G_{cp,z}$), in units of \$/MW-day, multiplied by the operating capacity of the unit ($O_u$) and the number of days in the month ($n$):

$$C_{m,u} = G_{cp,z} \times O_u \times n \quad (3)$$

Capacity market revenues are calculated for every coal-fired electricity generation unit in New England, New York, MISO, and PJM, and for every month in the period from March 1, 2020 to December 31, 2022.

*Overall profitability for the period between March 1, 2020 and December 31, 2022.* The overall profitability of each coal-fired generation unit is estimated for the period between March 1, 2020 and December 31, 2022. The overall profitability ($P$) for each unit ($u$) is estimated as the sum of discounted cash flows in each month, where $m$ ranges from 1 to 34, the number of months in the period. $L_m$ is the monthly nominal net cash flow, and $r$ is the WACC (Eq. (4)). We apply a monthly WACC of 0.38% that we derive from an annual WACC of 4.61%, consistent with the NREL Annual Technology Baseline 2019.

$$P_u = \sum^{m} \frac{L_m}{(1+r)^m} \quad (4)$$

The monthly nominal net cash flow ($L_m$) is the sum of monthly nominal electricity and capacity market revenues net of the sum of monthly nominal variable costs and fixed O&M costs. The profitability of every coal generation unit is calculated in two scenarios: a counterfactual scenario that relies on an electricity market price forecast published by the EIA in January 2020 prior to shelter-in-place orders, and a scenario that reflects our current expectations and is based on actual prices in March through December, and a price forecast published by the EIA in January 2021.

**Reporting summary**. Further information on research design is available in the Nature Research Reporting Summary linked to this article.

## Data availability

Most of the data that support the findings of this study are publicly available via the following sources and web links. For net generation by fuel, refer to U.S. Energy Information Administration (EIA), Form EIA-923, available at https://www.eia.gov. For fuel code-specific emissions factors, refer to EIA, Carbon Dioxide Emissions Coefficients, available at http://www.eia.gov; EIA, Carbon Dioxide Uncontrolled Emission Factors, available at https://www.eia.gov; and U.S. Environmental Protection Agency (EPA), Emission Factors for Greenhouse Gas Inventories, available at https://www.epa.gov. For population-weighted heating degree days (HDDs) and cooling degree days (CDDs), refer to EIA, Short-Term Energy Outlook Data Browser, available at https://www.eia.gov. For forecasts of monthly average regional electricity market prices, refer to EIA, Short-Term Energy Outlook (January 2020), available at https://www.eia.gov; and EIA, Short-Term Energy Outlook (January 2021), available at https://www.eia.gov. For the weighted average cost of capital (WACC) of coal-fired electricity generation units, refer to U.S. National Renewable Energy Laboratory (NREL), Annual Technology Baseline 2019, available at https://data.nrel.gov. The exception is data furnished by S&P Global Market Intelligence, which was made available to Max Luke, a former employee of NERA Economic Consulting (NERA), via NERA's paid subscription to S&P Global Market Intelligence service.

## Code availability

Full open source Python code and Excel calculations to replicate the results presented in this paper are available via GitHub at the following URL: https://github.com/highlandenergy/no-covid-19-climate-silver-lining-in-the-us-power-sector[54].

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

## Acknowledgements
The authors would like to thank Dr. Jesse D. Jenkins and Dr. David Harrison for their support with this work. They would also like to thank Dr. Edward S. Rubin, Doug Houseman, Bruce Nilles, Laura Martin, Ted Nace, Dr. Christopher Greig, Dr. Christian Hauenstein, and Dr. Audun Botterud for providing expertise with respect to coal power plant investment decisions and capacity market bidding behavior.

## Author contributions

The authors worked collectively to design the study. S.L., P.S., and D.S. performed the GP regression data analysis with guidance from M.L. M.L. and P.S. developed the energy and capacity market profitability model with guidance from S.L. M.L., T.C., D.S., and S.L. drafted and edited the paper. All authors approved the submitted version.

## Competing interests

The authors declare no competing interests.
