## [Peer Review File · Nature Communications]

Reviewer comments, first round -

Reviewer #1 (Remarks to the Author):

This study used a Gaussian processes regression approach to estimate the impact of COVID-19 on CO2 emissions accounting for variability in emissions unrelated to the pandemic. The relationship between COVID-19 and CO2 emissions reductions is undoubtedly an important issue to offer scientific measures to solve. From this perspective, this study presented its value. However, some drawbacks are existed in the current form of this research. From my perspective, the main problems that I suggest to hope improve the quality of this paper are listed as follows.

Comment 1 :

In the Abstract section, the authors introduce the necessity and significance of this research. However, the abstract is too long, so it should be kept as short as possible while highlighting the core of the problem and the conclusion.

Comment 2 :

The author quoted references in all sections of the whole article, which is rather confusing and inconvenient for readers to understand the relevant theoretical basis and methodological system. In order to enhance the logic of relevant literature citations, it is recommended to add a separate literature review section.

Comment 3 :

In the section 2, the author's introduction of how to effectively combine different emission data is not detailed enough, which will make readers a little confused. Therefore, the specific details should be further clarified.

Comment 4 :

In the section 3, the author conducted an independent analysis from the perspectives of E and C/E to study the impact of COVID-19 on reducing carbon emissions. Did they ignore the correlation between the two? In addition, when discussing the influencing factors of E, the author rarely mentioned changes in other social factors, but only mentioned the overall rate of decline of GDP, and drawing the conclusion that GDP is the most important influencing factor and is not strict enough. Thus, it is recommended to increase the screening research on the main influencing factors to improve the scientific nature of the whole paper.

Comment 5 :

In the conclusion part, it is mainly the simple repetition of the research content of each part. The policy recommendations given by the author are not clear yet, and it is recommended to strengthen the relevant discussion.

Reviewer #2 (Remarks to the Author):

Review of "No COVID-19 Climate "Silver Lining" in the U.S. Power Sector: CO2 Emissions Reductions Not Statistically Significant, Additional Risk to Coal Generators is Minimal"

The main question of this paper is whether the COVID-19 pandemic, with the consequent restrictions on societal and industrial activities, will have any lasting effect on CO2 emissions and the number of profitable coal-fired power plants in the USA.

Three approaches are used. The first identifies a statistical relationship between time and CO2 emissions from electricity generation. The second uses a similar method to find the relationships between time and electricity generation, and time and the carbon intensity of electricity generation. The third uses a counterfactual scenario to determine whether coal-fired generator

profitability resulting from electricity prices forecast after the advent of the pandemic is different to that resulting from prices forecast before the pandemic.

This question and the methods used to address it are both novel and interesting. I do have a number of comments that should be addressed.

page 3, data:

The authors average data in the overlapping period of two data series. This introduces additional uncertainty into the analysis that does not appear to be mentioned or accounted for, nor is any potential explanation provided for the differences between the two datasets used.

page 3, methods

The method used is Gaussian process regression with "a linear kernel, a bias kernel, and a standard periodic kernel." For readers not familiar with the GP method, it might be simply to just say that the model uses a constant term, a trend, and a mixture of sinusoidal terms. The use of the term 'kernel' might be correct in the GP literature, but I suspect many readers will not be familiar with this. It might also be useful for the general reader to simply summarise GP: what differentiates it (I suppose) is the use of Bayesian approaches to both the model fitting and the uncertainty assessment. But the explanatory variables used (constant, trend, sinusoids) would be familiar to readers if alternative terms were used.

The authors capture average (climate) temperature effects using sinusoidal terms in the regression, but do not include temperature directly. Adding temperature would very likely improve the explanatory power of the model, thereby reducing uncertainty, and we well might then that the March-July period falls outside of the uncertainty range, flipping the conclusion to the exact opposite of what is found. Given that it is known that temperature variations are a strong driver of electricity demand, why have the authors not included this in their analysis?

page 4, results:

the training period of the model is only four years (48-50 data points). The authors should make some comment about the consequence of this short period on the reliability of the analysis, particularly given that temperature is not included in the model.

The authors state that "an overall decreasing trend in emissions is observed", but this can only be determined by eye from Figure 1. This conclusion should be obtained from model results directly, and presented in %/yr or similar.

page 4, discussion:

The authors suggest that population and demographic changes could have explanatory power not captured by the model, but it is difficult to see which population or demographic effects could operate in such a short time-frame. These normally influence demand as trends, not intra-annually. Please explain.

The way the authors cite reference 3 and 5 suggests the authors' analysis contradicts statements made by others. But the two articles cited here present conclusions for global and Chinese emissions, while the present study is constrained to the USA. I'd suggest rewording this.

The discussion of the January 2018 weather bomb reads as if the weather event caused a constraint in the gas pipeline, but that appears to be an incorrect reading of the source. With limited installed capacity of natural gas pipelines, and a reduction in other (presumably coal) generation options, the extreme cold and resultant very high electricity demand meant that other energy sources had to be used for generation, including oil. Please check the source, including the original article (https://www.eia.gov/special/alert/east_coast/) and reword as necessary. This is important, because it was the cold itself, not the weather's effects on infrastructure, that caused this surge in electricity emissions. Emissions would have spiked without the pipeline constraint, but spiked higher because of it. If temperature were included in the model, this would have been captured.

A simple statement about the (ab)normality of temperatures/weather during the COVID-19 period

of study is necessary here to conclude that the COVID-19 deviations are not unprecedented.

page 5

Strictly speaking, at time writing the US has only announced its intention to withdraw, but this withdrawal will occur on Nov 4, 2020, before publication.

"If the 5.6% observed average reduction in CO2 emissions concurrent with COVID-19 were to persist..."

I have difficulty accepting this statement when the authors have already concluded that these reductions are not statistically significant. In effect, one cannot say with confidence from the analysis that they have occurred. Two paragraphs earlier they explicitly call into question a causal link between COVID-19 and an emissions decline. What reason, then, would there be to consider a persistence in this short-term reduction? Moreover, if this analysis were performed only a few months' later, it would be somewhat clearer whether this is a trend or not. The authors provide no analysis to show why emissions dropped concurrent with COVID-19, so any supposition about continuance of this drop needs much more careful couching in terms of scenario language. For example, presented as a hypothetical situation. But even then, the the hypothesis is only relevant in the context of the authors suggesting their conclusion is incorrect ("Even if the changes were significant, ...").

page 6, discussion

Top of second column: "reduction in the energy intensity" should presumably read "reduction in the electricity intensity".

The authors use the highly simplified indicator of GDP in this analysis and discussion. In reality an increase in economic activity in some industries combined with a decrease in others might result in no change in GDP but a substantial change in electricity demand. This should be discussed.

page 7

I appreciate that there might not be readily available literature specifically on the link between teleworking and electricity consumption, but the present article is specifically about electricity consumption, not energy use more broadly, and this sentence and the citation used fall outside of the scope of the paper, which is electricity. The citation does not appear to support the argument that there is no link between teleworking and electricity consumption.

"widespread electric vehicle automation":

Given the previous discussion takes the reader's mind to factory automation, confusion would be reduced here by referring instead to "widespread introduction of autonomous electric vehicles." I had to check the titles of the cited articles to understand that this was the meaning here.

"Those forecasts reflect the average expectations of more than 70 Wall Street Journal economists [49]":

I don't believe this is correct. The website appears to have reported a survey performed by the Wall Street Journal of more than 70 economists. This is not the expectation of the American Staffing Association, and nor are these "Wall Street Journal economists." Please reword.

<https://web.archive.org/web/20200716180519/https://americanstaffing.net/staffing-research-data/asa-data-dashboard/gdp-quarterly-projections/>

page 8, section 3.4.2

This discussion section only presents results without interpretation, before introducing what the next section will address. This section should reiterate the possibility that C/E declined because of merit-order dispatch (relative prices).

page 9

"Two electricity market price scenarios are constructed.":

Are they "constructed" or simply taken directly from the EIA? If there's more to it than taking them from the EIA (capacity weighting) then this is unclear.

page 12

Given how difficult short-term forecasting is at the moment, the statements about EIA expectations should be updated to use the latest STEOs before publication. The manuscript as reviewed refers to the July STEO, while the current at time of review is the October STEO, and there have already been significant changes in that period. In fact the October STEO forecasts that coal generation will be 12% higher in 2021 than they forecast in the January STEO. The reason for this is an expected increase in natural gas prices

page 13

"A reduction in GDP is responsible in large part for the overall reduction in E.":

More correctly, a reduction in activity in the real economy is responsible for both a drop in GDP and a drop in electricity demand.

"Our results suggest that COVID-19 led to reductions in power sector CO2 emissions in the contiguous U.S.; however, they also suggest that the magnitudes of those reductions are not as obvious as several studies suggest.":

The analysis purports to show that there are no significant changes, and if the authors trust their own model then they cannot say that the deviations are caused by COVID-19.

"Within a couple of years, CO2 emissions are likely to increase to levels we would have expected prior to the pandemic.":

The authors should maintain the scope of any conclusions to the electricity sector. If nothing else changes in the USA because of COVID-19, then the analysis here (assuming it is correct) demonstrates that CO2 emissions from electricity generation would return to trend. Obviously other things have changed, so the language here needs to be careful. While the direct effects of societal constraints on electricity demand might be short-lived, other indirect effects are likely in play.

Reviewer #3 (Remarks to the Author):

This is a relatively well-written article on the COVID-19 impact on US power sector with emphasis on CO2 emissions and coal-based electricity generation. Below are some specific comments for the authors:

1. While majority of the articles written about COVID-19 and electricity reported reduced demand, there are some articles that do not conclude the same (at least report regional variability of the COVID-19 impact). There might be some merit to reviewing these articles:

- López Prol, Javier, and Sungmin O. 2020. "Impact of COVID-19 Measures on Short-Term Electricity Consumption in the Most Affected EU Countries and USA States." *iScience* 23 (10): 101639.

- Agdas, D., and P. Barooah. 2020. "Impact of the COVID-19 Pandemic on the US Electricity Demand and Supply: An Early View from Data." *IEEE Access*.
<https://ieeexplore.ieee.org/abstract/document/9169615/>.

2. Paragraph 3 on page 2 can be shortened. The counterfactual demand term is commonly used.

3. Paragraph 6 on page 2 needs to be elaborated better. The argument of GDP related electricity generation is plausible but this also assumes the consumption trends are unaffected by the pandemic. There is EIA analyses that report reduced commercial consumption that is offset by increased residential consumption in parts of US (<https://www.eia.gov/todayinenergy/detail.php?id=43636>). I would like to see additional justification here.

4. The flow of the article is subpar as is. The introduction section is hard to follow. The authors introduce a number of concepts but none are sufficiently described (see comment 3).

5. Similarly, I am having difficulty seeing how coal plant profitability fits to this article from a scope perspective. Is the potential of coal plant retirement or early retirement because of lack of profitability part of the silver lining?

6. First paragraph in section 2 is redundant. These statements already appear in the first section in the first page.

7. I would like to see more on the GP regression development details. The authors appear to rely on these models in substantiating their claims about statistical assessment of CO2 emission reduction but the model development details are virtually non-existent. I would like to know more about how GP regression works, details of model development criteria and so on. For instance, what is L-BFGS-B algorithm, why 5 random restarts was chosen by the authors? While the justification for using GP regression based on literature review is sufficient, more details need to be presented here.

8. What is credible interval (first appears in Figure 2)? Is this the same as confidence interval, and if so, how was this calculated?

9. Paragraph 2 in page 4 needs to be rethought. There is no hypothesis testing done here, it is simply shown that the deviations are within a credible/confidence interval.

10. Similarly, how were these intervals estimated?

11. While I agree with the statement in paragraph 3 of page 4 about the causality between COVID19 and CO2 reductions, I don't think the authors provide sufficient discussions for a robust rebuttal. I don't believe the variability induced by different factors listed here (weather, demographic etc) should be combined together.

12. In section 3, the authors do not present a hypothesis testing, but simply compare observed numbers to predicted ones (Also see comment 9).

13. I don't find the arguments in paragraph 2 on page 7 convincing. The authors over-rely on literature and perhaps under-analyse data here. For instance, authors comment on the impact on teleworking on energy demand--although the citation they rely on seems to be pre-COVID. There is ample data and reports available (see comment 3) to have a more granular assessment of this. This is a very important part of the article and it is not treated as such.

14. The conclusions are weak. They are essentially a summary of the article and the only conclusion unique to this article is the claim that CO2 emissions are likely to increase to levels anticipated prior to the pandemic in the next couple of years. I believe this is contradictory to the authors' own statement on the first paragraph on page 8. Here, the authors claim that electricity generation should be close to expected levels in the absence of the pandemic because of the predicted GDP increase anticipated in Q3--note that the actual GDP growth far exceeded this prediction. So, can we deduct the CO2 levels are back or higher than a no-pandemic level right now because of the increased electricity generation? This might be a wording issue, but it is important and should be clarified.

Overall, I think there is merit in what is proposed here but I believe the emphasis of the analysis and presentation should be reconsidered.

Ref	Reviewer Comment	Authors' Response	Type
R1: #1	In the Abstract section, the authors introduce the necessity and significance of this research. However, the abstract is too long, so it should be kept as short as possible while highlighting the core of the problem and the conclusion.	We have considerably shortened the Abstract.	CHANGE MADE
R1: #2	The author quoted references in all sections of the whole article, which is rather confusing and inconvenient for readers to understand the relevant theoretical basis and methodological system. In order to enhance the logic of relevant literature citations, it is recommended to add a separate literature review section.	We have attempted to streamline paper such that most of the non-methodological references appear either in the Introduction section or the Discussion section. We have aggregated the Discussion section into a single section.	CHANGE MADE
R1: #3	In the section 2, the author's introduction of how to effectively combine different emission data is not detailed enough, which will make readers a little confused. Therefore, the specific details should be further clarified.	We have changed our methodology and documentation to reflect how we are now only using one source of generation data. Form EIA-923, and the step of converting this to emissions data is documented: multiplying generation data by EIA and EPA-provided fuel-specific factors.	CHANGE MADE
R1: #4	In the section 3, the author conducted an independent analysis from the perspectives of E and C/E to study the impact of COVID-19 on reducing carbon emissions. Did they ignore the correlation between the two? In addition, when discussing the influencing factors of E, the author rarely mentioned changes in other social factors, but only mentioned the overall rate of decline of GDP, and drawing the conclusion that GDP is the most important influencing factor and is not strict enough. Thus, it is recommended to increase the screening research on the main influencing factors to improve the scientific nature of the whole paper.	This note is well-taken. As R1 points out, correlation between E and C/E is important (and can actually be visually appreciated by comparing Fig. 2 and 3). We discuss E and (C/E) independently because the equation $C = E * (C/E)$ is valid regardless of correlation between the E and C/E terms. We argue that if trends in E are likely to persist, and if trends in C/E are likely to persist, then trends in C are likely to persist. We augment our GDP argument in the Discussion section to include literature about how COVID-19 affects E: "For	CHANGE MADE

		example, in a recent report [13] researchers conclude that COVID-19-related shelter-in-place orders could trigger a sustained long-term reduction in U.S. electricity demand of 65-160 terawatt-hours, or 1.6-4.0% of annual electricity demand. In their central scenario, the authors estimate a transition of about 11% of U.S. office workers to permanent work-from-home positions, permanent decreases in office and retail-related electricity consumption, and a less-impactful permanent increase in residential electricity consumption.”	
R1: #5	In the conclusion part, it is mainly the simple repetition of the research content of each part. The policy recommendations given by the author are not clear yet, and it is recommended to strengthen the relevant discussion.	We have completely revised the Discussion and Conclusion sections in a way that we believe addresses this point. The Conclusion is no longer a restatement of our results.	CHANGE MADE
R2: #1	page 3, data: The authors average data in the overlapping period of two data series. This introduces additional uncertainty into the analysis that does not appear to be mentioned or accounted for, nor is any potential explanation provided for the differences between the two datasets used.	We have changed our methodology and documentation to reflect how we are now only using one source of generation data: Form EIA-923. We believe this addresses this issue.	CHANGE MADE
R2: #2	page 3, methods: The method used is Gaussian process regression with "a linear kernel, a bias kernel, and a standard periodic kernel." For readers not familiar with the GP method, it might be simply to just say that the model uses a constant term, a trend, and a mixture of sinusoidal terms. The use of the term 'kernel' might be correct in the GP literature, but I suspect many readers will not be familiar with this.	This note is well-taken. We have decided to include both vocabularies, to communicate with a more general audience while staying more precise. We have also expounded upon how we justify our use of GPs: “GPs represent a class of Bayesian nonparametric models. They assume that every finite collection of random variables has a multivariate normal distribution. GP	CHANGE MADE

	It might also be useful for the general reader to simply summarise GP: what differentiates it (I suppose) is the use of Bayesian approaches to both the model fitting and the uncertainty assessment. But the explanatory variables used (constant, trend, sinusoids) would be familiar to readers if alternative terms were used.	regressions can be used, as in our case, to forecast the likely ranges of variables based on the historical distributions of those variables. GP regressions are appropriate for testing the statistical significance and estimating the uncertainty of outcomes impacted by seasons and other periodic variables.”	
R2: #3	The authors capture average (climate) temperature effects using sinusoidal terms in the regression, but do not include temperature directly. Adding temperature would very likely improve the explanatory power of the model, thereby reducing uncertainty, and we well might then that the March-July period falls outside of the uncertainty range, flipping the conclusion to the exact opposite of what is found. Given that it is known that temperature variations are a strong driver of electricity demand, why have the authors not included this in their analysis?	We have enhanced our GP methodology to include heating degree day (HDD) and cooling degree day (CDD) data for the contiguous U.S. explicitly as a way of incorporating temperature. We appreciate R2 for emphasizing the importance of doing this.	CHANGE MADE
R2: #4	page 4, results: the training period of the model is only four years (48-50 data points). The authors should make some comment about the consequence of this short period on the reliability of the analysis, particularly given that temperature is not included in the model.	We have expanded upon our justification for this date range: “This historical period is long enough to obtain a strong model fit but not so long as to necessitate additional nonlinear approximations or more complicated multiyear regression methods.” As stated previously, we are now incorporating temperature via HDD and CDD data.	CHANGE MADE
R2: #5	The authors state that "an overall decreasing trend in emissions is observed", but this can only be determined by eye from Figure 1. This conclusion should be obtained from model results directly, and presented in %/yr or similar.	We decided to remove this statement because, while it is true, it is not directly material to our conclusions. Changes to our GP model to include HDD and CDD data also makes this is visible within Fig. 1, so we don't want to distract our readers with	CHANGE MADE

		this statement.	
R2: #6	page 4, discussion: The authors suggest that population and demographic changes could have explanatory power not captured by the model, but it is difficult to see which population or demographic effects could operate in such a short time-frame. These normally influence demand as trends, not intra-annually. Please explain.	We thank R2 for pointing out this potential overstatement of the impacts of intra-annual population and demographic trends. We now try to provide better examples of impactful factors that are ignored in modeling: “Such factors associated with this variability may include short-term macroeconomic changes, discrete or non-linear changes to the power-system, and outlier events.”	CHANGE MADE
R2: #7	The way the authors cite reference 3 and 5 suggests the authors’ analysis contradicts statements made by others. But the two articles cited here present conclusions for global and Chinese emissions, while the present study is constrained to the USA. I'd suggest rewording this.	We have reworded the relevant sentence in such a way that we believe removes the contradiction. Thank you for identifying this.	CHANGE MADE
R2: #8	The discussion of the January 2018 weather bomb reads as if the weather event caused a constraint in the gas pipeline, but that appears to be an incorrect reading of the source. With limited installed capacity of natural gas pipelines, and a reduction in other (presumably coal) generation options, the extreme cold and resultant very high electricity demand meant that other energy sources had to be used for generation, including oil. Please check the source, including the original article (https://www.eia.gov/special/alert/east_coast/) and reword as necessary. This is important, because it was the cold itself, not the weather's effects on infrastructure, that caused this surge in electricity emissions. Emissions would have spiked without the pipeline constraint, but spiked higher because of it. If temperature were included in the model, this would have been captured.	You are correct. We have amended our description of the “bomb cyclone” event accordingly.	CHANGE MADE

R2: #9	A simple statement about the (ab)normality of temperatures/weather during the COVID-19 period of study is necessary here to conclude that the COVID-19 deviations are not unprecedented.	We decided to remove the discussion about the (ab)normality of temperatures/weather during the COVID-19 study period. We believe this a sound decision because we now explicitly model temperature/weather by way of HDDs and CDDs.	CHANGE MADE
R2: #10	page 5: Strictly speaking, at time writing the US has only announced its intention to withdraw, but this withdrawal will occur on Nov 4, 2020, before publication.	We decided to remove this discussion altogether because we do not believe that it enhances the discussion or the conclusions of the paper.	CHANGE MADE
R2: #11	"If the 5.6% observed average reduction in CO2 emissions concurrent with COVID-19 were to persist..." I have difficulty accepting this statement when the authors have already concluded that these reductions are not statistically significant. In effect, one cannot say with confidence from the analysis that they have occurred. Two paragraphs earlier they explicitly call into question a causal link between COVID-19 and an emissions decline. What reason, then, would there be to consider a persistence in this short-term reduction? Moreover, if this analysis were performed only a few months' later, it would be somewhat clearer whether this is a trend or not. The authors provide no analysis to show why emissions dropped concurrent with COVID-19, so any supposition about continuance of this drop needs much more careful couching in terms of scenario language. For example, presented as a hypothetical situation. But even then, the the hypothesis is only relevant in the context of the authors suggesting their conclusion is incorrect ("Even if the changes were significant, ...").	We agree. We decided to remove this discussion altogether because we do not believe that it enhances the discussion or the conclusions of the paper. We have also scrutinized and revised our language regarding statements about the possible continuance (or not) of COVID-19-related reductions in CO2 emissions.	CHANGE MADE
R2: #12	page 6, discussion: Top of second column: "reduction in the energy intensity" should presumably read "reduction in the electricity intensity".	Thank you. We have made the appropriate correction.	CHANGE MADE

R2: #13	The authors use the highly simplified indicator of GDP in this analysis and discussion. In reality an increase in economic activity in some industries combined with a decrease in others might result in no change in GDP but a substantial change in electricity demand. This should be discussed.	Excellent point. We have amended the discussion in such a way that we believe addresses this point.	CHANGE MADE
R2: #14	page 7: I appreciate that there might not be readily available literature specifically on the link between teleworking and electricity consumption, but the present article is specifically about electricity consumption, not energy use more broadly, and this sentence and the citation used fall outside of the scope of the paper, which is electricity. The citation does not appear to support the argument that there is no link between teleworking and electricity consumption.	We agree and we have removed the reference to the teleworking study. You are correct that it did not contribute to the paper.	CHANGE MADE
R2: #15	"widespread electric vehicle automation": Given the previous discussion takes the reader's mind to factory automation, confusion would be reduced here by referring instead to "widespread introduction of autonomous electric vehicles." I had to check the titles of the cited articles to understand that this was the meaning here.	Thank you. We decided to remove references to electric vehicle automation and instead focus on automation more generally. We believe this addresses your point.	CHANGE MADE
R2: #16	"Those forecasts reflect the average expectations of more than 70 Wall Street Journal economists [49]": I don't believe this is correct. The website appears to have reported a survey performed by the Wall Street Journal of more than 70 economists. This is not the expectation of the American Staffing Association, and nor are these "Wall Street Journal economists." Please reword. https://web.archive.org/web/20200716180519/https://americanstaffing.net/staffing-research-data/asa-data-dashboard/gdp-quarterly-projections/	You are correct. We decided to remove reference to the "average expectations of more than 70 Wall Street Journal economists" and to the ASA, and instead to add references to economic forecasts conducted by the Federal Reserve Board, Congressional Budget Office, OECD, and others.	CHANGE MADE
R2: #17	page 8, section 3.4.2: This discussion section only presents results without interpretation, before introducing what the next section will address. This section should reiterate the	We decided to remove that section (3.4.2) because, as you correctly pointed out, it presented results without interpretation. We	CHANGE MADE

	possibility that C/E declined because of merit-order dispatch (relative prices).	have streamlined the document such that all of the discussion resides in a single Discussion section.	
R2: #18	page 9: "Two electricity market price scenarios are constructed.": Are they "constructed" or simply taken directly from the EIA? If there's more to it than taking them from the EIA (capacity weighting) then this is unclear.	The hourly price scenarios are constructed using monthly price scenarios published by the EIA. The hourly price scenarios, which are derived from the monthly EIA price scenarios, are used in our estimates of coal plant profitability. We have attempted to explain this more clearly in the manuscript.	CHANGE MADE
R2: #19	page 12: Given how difficult short-term forecasting is at the moment, the statements about EIA expectations should be updated to use the latest STEOs before publication. The manuscript as reviewed refers to the July STEO, while the current at time of review is the October STEO, and there have already been significant changes in that period. In fact the October STEO forecasts that coal generation will be 12% higher in 2021 than they forecast in the January STEO. The reason for this is an expected increase in natural gas prices	We agree completely. We have updated the coal profitability analysis to reflect the EIA's price forecast as published in January 2021.	CHANGE MADE
R2: #20	page 13: "A reduction in GDP is responsible in large part for the overall reduction in E.": More correctly, a reduction in activity in the real economy is responsible for both a drop in GDP and a drop in electricity demand.	Thank you, your point is well taken. We have amended the language to distinguish between activity in the real economy and its relationships to GDP and electricity demand.	CHANGE MADE
R2: #21	"Our results suggest that COVID-19 led to reductions in power sector CO2 emissions in the contiguous U.S.; however, they also suggest that the magnitudes of those reductions are not as obvious as several studies suggest.": The analysis purports to show that there are no significant changes, and if the authors trust their own model then they cannot say that the deviations are caused by COVID-19.	You are absolutely right. We have revised the text such that we do not contradict the results of our analysis, which we trust.	CHANGE MADE

R2: #22	"Within a couple of years, CO2 emissions are likely to increase to levels we would have expected prior to the pandemic.": The authors should maintain the scope of any conclusions to the electricity sector. If nothing else changes in the USA because of COVID-19, then the analysis here (assuming it is correct) demonstrates that CO2 emissions from electricity generation would return to trend. Obviously other things have changed, so the language here needs to be careful. While the direct effects of societal constraints on electricity demand might be short-lived, other indirect effects are likely in play.	We agree. We have amended the language to ensure that our discussion and conclusions stay within the scope of the analysis (US power sector CO2 emissions). We have also attempted to take more care in our discussion of the potential long-term impacts of COVID-19 on US power sector CO2 emissions. We believe that our changes address your point.	CHANGE MADE
R3: #1	While majority of the articles written about COVID-19 and electricity reported reduced demand, there are some articles that do not conclude the same (at least report regional variability of the COVID-19 impact). There might be some merit to reviewing these articles:  - López Prol, Javier, and Sungmin O. 2020. "Impact of COVID-19 Measures on Short-Term Electricity Consumption in the Most Affected EU Countries and USA States." iScience 23 (10): 101639. - Agdas, D., and P. Barooah. 2020. "Impact of the COVID-19 Pandemic on the US Electricity Demand and Supply: An Early View from Data." IEEE Access. https://ieeexplore.ieee.org/abstract/document/9169615/. 	Thank you. We have reviewed these articles and included them in our discussion of the literature. We have also reviewed and included other studies that we either missed or that were published since our last submission. Those include:  • Shan et al., "Impacts of COVID-19 and fiscal stimuli on global emissions and the Paris Agreement," Nature Climate Change 11, 200-206 (2020) • Le Quéré et al., "Fossil CO2 emissions in the post-COVID-19 era," Nature Climate Change 11, 197-199 (2021) 	CHANGE MADE
R3: #2	Paragraph 3 on page 2 can be shortened. The counterfactual demand term is commonly used.	Thank you, we agree. We have significantly amended and shortened the discussion about causality.	CHANGE MADE
R3: #3	Paragraph 6 on page 2 needs to be elaborated better. The argument of GDP related electricity generation is plausible but this also assumes the consumption trends are unaffected by the pandemic. There is EIA analyses that report reduced	We have decided to reserve our discussion of GDP to the Discussion section, and not to address it in earlier sections because GDP is not the focus of our analysis. In the	CHANGE MADE

	commercial consumption that is offset by increased residential consumption in parts of US (https://www.eia.gov/todayinenergy/detail.php?id=43636). I would like to see additional justification here.	Discussion section, we have written a more nuanced description of the possibility of an increase in economic activity but changes in the industrial and sectoral patterns of electricity demand and CO2 emissions.	
R3: #4	The flow of the article is subpar as is. The introduction section is hard to follow. The authors introduce a number of concepts but none are sufficiently described (see comment 3).	We have significantly revised the structure of the paper in such a way that enhances flow and readability. We hope that you agree.	CHANGE MADE
R3: #5	Similarly, I am having difficulty seeing how coal plant profitability fits to this article from a scope perspective. Is the potential of coal plant retirement or early retirement because of lack of profitability part of the silver lining?	We believe that an analysis of coal plant profitability is warranted because coal power generation accounts a larger share of US power sector CO2 emissions than any other source and because coal generation and CO2 emissions were lower on average in COVID-19 months than other fossil fuel generation sources (as we report in Section 3). If we had found that COVID-19 was likely to lead to a large share of coal plant retirements (due to a lack of profitability from depressed power prices) then we might have concluded that COVID-19 does indeed have a climate “silver lining”.	CHANGE MADE
R3: #6	First paragraph in section 2 is redundant. These statements already appear in the first section in the first page.	We are now omitting this redundant section.	CHANGE MADE
R3: #7	I would like to see more on the GP regression development details. The authors appear to rely on these models in substantiating their claims about statistical assessment of CO2 emission reduction but the model development details are virtually non-existent. I would like to know more about how GP regression works, details of model development	This note is well-taken. We expand upon our description of GPs by stating: “GP regression models are a class of Bayesian nonparametric models. They assume that every finite collection of random variables has a multivariate normal distribution.”	CHANGE MADE

	criteria and so on. For instance, what is L-BFGS-B algorithm, why 5 random restarts was chosen by the authors? While the justification for using GP regression based on literature review is sufficient, more details need to be presented here.	While the mechanics of GP regression are fairly technical in nature, they are well-documented and widely-used. We provide additional references for their technical descriptions and for related applications. We improve our justification of our GP fitting routine as requested: “The GP model is fit using a SciPy implementation of the L-BFGS-B algorithm with five random restarts [ref]. The L-BFGS-B algorithm is a standard optimization technique [ref]: L-BFGS-B and multiple random restarts are commonly used to fit GPs. Our optimization settings yield model runs that are highly stable and reproducible.”	
R3: #8	What is credible interval (first appears in Figure 2)? Is this the same as confidence interval, and if so, how was this calculated?	We now updated our terminology to only use the more common term, “confidence interval”. Credible intervals and confidence intervals are synonymous given our use of GPs and the Bayesian modeling paradigm. We also now describe in our paper how this is calculated using the Gaussian distributions our models characterize after we fix time, HDD, and CDD data points. A 95% CI is given as the range spanning +/-1.96 standard deviations on either side of the GP’s mean forecast, as is standard for Gaussian distributions.	CHANGE MADE
R3: #9	Paragraph 2 in page 4 needs to be rethought. There is no hypothesis testing done here, it is simply shown that the deviations are within a credible/confidence interval.	We now clarify our hypothesis testing routine via a simple decision rule: “Given these Gaussian distributions, statistical hypothesis testing amounts to determinations of 95% confidence intervals	CHANGE MADE

		(CIs) and a simple decision rule: if the observed CO2 emissions level is outside of the 95% CI, we reject the null hypothesis that the observation could have occurred with reasonable probability in the absence of COVID-19. In other words, if the observed CO2 emissions level is outside of the 95% CI, we infer statistically significant impacts of COVID-19 on CO2 emissions. If the observed value is within the 95% CI, we accept or 'fail to reject' the null hypothesis and conclude that deviations are not statistically significant under our decision rule."	
R3: #10	Similarly, how were these intervals estimated?	We now describe in our paper how this is calculated using the Gaussian distributions the model characterizes once we fix time, HDD, and CDD, as described in the model, as quoted in the cell above.	CHANGE MADE
R3: #11	While I agree with the statement in paragraph 3 of page 4 about the causality between COVID19 and CO2 reductions, I don't think the authors provide sufficient discussions for a robust rebuttal. I don't believe the variability induced by different factors listed here (weather, demographic etc) should be combined together.	We have now updated and enhanced our modeling to now explicitly account for weather via heating degree days (HDD) and cooling degree days (CDD). We provide documentation of these changes in the paper. All other variability is accounted for as Gaussian noise, and is only modeled implicitly through our characterizations of uncertainty.	CHANGE MADE
R3: #12	In section 3, the authors do not present a hypothesis testing, but simply compare observed numbers to predicted ones (Also see comment 9).	We now clarify our hypothesis testing routine via a simple decision rule, as expanded upon in our response to comment 9.	CHANGE MADE

R3: #13	I don't find the arguments in paragraph 2 on page 7 convincing. The authors over rely on literature and perhaps under analyse data here. For instance, authors comment on the impact on teleworking on energy demand--although the citation they rely on seems to be pre-COVID. There is ample data and reports available (see comment 3) to have a more granular assessment of this. This is a very important part of the article and it is not treated as such.	We agree and have decided to restructure our discussion in a way that we believe to be more convincing. We have removed reference to the teleworking study because we do not believe that it contributes to the discussion.	CHANGE MADE
R3: #14	The conclusions are weak. They are essentially a summary of the article and the only conclusion unique to this article is the claim that CO2 emissions are likely to increase to levels anticipated prior to the pandemic in the next couple of years. I believe this is contradictory to the authors own statement on the first paragraph on page 8. Here, the authors claim that electricity generation should be close to expected levels in the absence of the pandemic because of the predicted GDP increase anticipated in Q3--note that the actual GDP growth far exceeded this prediction. So, can we deduct the CO2 levels are back or higher than a no-pandemic levels right now because of the increased electricity generation? This might be a wording issue, but it is important and should be clarified.	Thank you. We have significantly amended the Discussion and Conclusion sections. We believe that the discussion and conclusion are much stronger. We no longer summarize the findings of the paper in the Discussion or Conclusion sections. We have attempted to remedy all instances in which we contradict ourselves.	CHANGE MADE
R3: #15	Overall, I think there is merit in what is proposed here but I believe the emphasis of the analysis and presentation should be reconsidered.	We greatly appreciate your comments. We have attempted to improve the analysis and presentation.	CHANGE MADE

Reviewer comments, second round –

Reviewer #1 (Remarks to the Author):

The authors have well addressed all of my concerns raised last time, I have no further comments.

Reviewer #2 (Remarks to the Author):

The changes made to the paper in this revision have improved clarity and the conclusions are better founded in the analysis. This is a nice piece of research.

Ref	Reviewer Comment	Authors' Response
Reviewer #1	The authors have well addressed all of my concerns raised last time, I have no further comments.	Thank you very much Reviewer #1.
Reviewer #2	The changes made to the paper in this revision have improved clarity and the conclusions are better founded in the analysis. This is a nice piece of research.	Thank you very much Reviewer #2.